# Palm Kernel Cake in Diets for Lactating Goats: Intake, Digestibility, Feeding Behavior, Milk Production, and Nitrogen Metabolism

**DOI:** 10.3390/ani12182323

**Published:** 2022-09-07

**Authors:** Fernanda G. Ferreira, Laudí C. Leite, Henry D. R. Alba, Douglas dos S. Pina, Stefanie A. Santos, Manuela S. L. Tosto, Carlindo S. Rodrigues, Dorgival M. de Lima Júnior, Juliana S. de Oliveira, José E. de Freitas Júnior, Bruna M. A. de C. Mesquita, Gleidson G. P. de Carvalho

**Affiliations:** 1Department of Animal Science, Federal University of Bahia, Av. Adhemar de Barros, 500, Ondina, Salvador 40170110, Brazil; 2Department of Animal Science, Universidade Federal do Recôncavo da Bahia, Cruz das Almas 44380000, Brazil; 3Department of Animal Science, Universidade Federal Rural do Semi-Árido, Mossoró 59625300, Brazil; 4Department of Animal Science, Universidade Federal da Paraíba, Areia 58397000, Brazil; 5Institute of Agricultural Sciences, Universidade Federal de Minas Gerais, Montes Claros 39404547, Brazil

**Keywords:** by-product, goat, milk production efficiency, palm kernel cake, performance, small ruminant

## Abstract

**Simple Summary:**

Palm kernel cake (PKC) is a by-product used in the diet of ruminants (sheep, dairy cows, beef cattle) with the aim of reducing production costs. The inclusion of PKC showed promising results mainly in feedlot animals. However, among domestic ruminant species, goats are the most selective feed animals, which may result in different feeding behavior and performance when PKC is included in the diet. We tested the dietary inclusion of PKC, a by-product of the biofuel industry, at the levels of 0, 80, 160 and 240 g kg^−1^, to evaluate its impact on the performance, feeding behavior and the metabolic profile of lactating goats. The inclusion of up to 80 g kg^−1^ PKC is recommended for the diet of goats without affecting the performance.

**Abstract:**

This study was carried out to determine the optimal inclusion rate of palm kernel cake (PKC) in diets for lactating goats based on intake, digestibility, feeding behavior, milk production and nitrogen metabolism. Twelve goats were used, eight Saanen and four Anglo Nubian, with a body weight of 46.0 ± 9.4 kg and 105 ± 5 days of lactation, distributed in a 4 × 4 Latin square design in triplicate. The diets were composed of increasing levels of PKC; 0, 80, 160 and 240 g kg^−1^ of dry matter (DM). The DM, crude protein, neutral detergent fiber and non-fibrous carbohydrates intakes decreased (*p* < 0.01) with the PKC inclusion. Milk production and milk protein decreased (*p* = 0.001), and milk production efficiency was not affected (*p* > 0.05) by the PKC inclusion. The ether extract intake varied quadratically (*p* < 0.01). Feeding and rumination efficiencies of dry matter and NDF were reduced (*p* < 0.001). The fat and total solids contents of the milk were influenced quadratically (*p* < 0.05). Nitrogen excretion (feces and milk), and retained nitrogen decreased (*p* < 0.001). Moderate use of up to 80 g kg^−1^ DM of PKC in the diet of lactating goats is recommended since at these levels milk production or body weight are significantly affected.

## 1. Introduction

It is estimated that by 2050, the world population will grow to 10 billion people [1]. In this context, it is imperative that animal productivity and production be more efficient to meet the food needs of the population. To meet the need for milk and milk products, we must adhere to technologies that increase production and continuity of supply, without affecting animal welfare. Confinement is a strategy that promotes increased production and product homogeneity; however, confinement generates an increase in production costs, with feed being the most expensive factor within this production system [2].

As a strategy to reduce feed costs, the use of by-products with adequate nutritional value for animal nutrition and lower commercial value than the concentrates traditionally used, mainly corn and soybean meal, has been studied [3]. According to Lisboa et al. [4], it is important to determine the optimal inclusion rate of the by-products, because these must be used with care to avoid unwanted effects in the production system. Using residues and by-products from vegetable oils agro-industry as ruminant feedstuffs for their recovery and valorization through circular-economy models (valorization strategy) promotes sustainable production [5]. The nutritional potential of the by-products can be determined from the evaluation of the intake, digestibility, feeding behavior, milk production and metabolic profile of the animals studied.

Palm kernel (*Elaeis guineensis*) cake (PKC) is a by-product of the biofuel industry, obtained after extracting the oil from the fruit, with regular supply throughout the year [6]. Its composition is favorable for its use in animal feed, since it has, on average, 157 g kg^−1^ of crude protein (CP), 548 g kg^−1^ of neutral detergent fiber (NDF) and 70 g kg^−1^ of ether extract (EE) on dry matter (DM) basis [7,8].

In the literature, it has been shown that the inclusion of PKC in the diet of feedlot goats up to the level of 210 g kg^−1^ DM did not affect DM intake [9]. Furthermore, it was observed that the optimal inclusion rate of PKC was 108 g kg^−1^ DM in high-concentrate diets for feedlot culling goats [7]. PKC is also recommended in the diet of feedlot beef cattle at an inclusion rate of up to 240 g kg^−1^ DM of the total diet [10].

The use of PKC in beef animals is promising [7,8,9,10]; however, the PKC use in dairy goat diets is too limited to make decisions about this by-product. Furthermore, it is important to consider that goats have specific characteristics and feeding behaviors that can influence the intake of different feed sources [11]. In view of the nutritional characteristics of PKC, as well as the results observed in the literature, our hypothesis is that there is an optimal rate of inclusion of PKC for lactating goats, which can improve milk production, without affecting negatively the intake and allowing reducing the production costs.

Therefore, the objective was to determine the optimal inclusion rate of palm kernel cake in diets for lactating goats based on intake, digestibility, feeding behavior, milk production and nitrogen metabolism.

## 2. Materials and Methods

### 2.1. Ethics Committee and Experiment Location

The experiment followed animal welfare rules, hence, the project was approved (approval no. 73/2018) by the Ethics Committee on the Use of Animals (CEUA) at the Federal University of Bahia (UFBA). The experiment was conducted in the goat farming section of UFBA, located in the municipality of Entre Rios—BA, Brazil (11°56′31″ S, 38°05′04″ W, 162 m above sea level).

### 2.2. Animals, Experimental Design and Management

Twelve multiparous lactating goats were used in a 4 × 4 Latin square design (four goat of the same breed, four periods and four treatments) in triplicate (three parallel Latin Squares) consisting of eight Saanen goats and four Anglo Nubian goats (multiparous, average weight of 46.9 ± 9.4, averaging 105 ± 5 days in milk; with an average production of 1.5 ± 0.4 kg day^−1^).

The study lasted 71 days, which included 15 days for the animals to acclimate themselves to the facilities, milking management and diets and 56 days divided into four experimental periods of 14 days. Each experimental period was composed of ten days to adapt animals to the treatments and four days for data collection.

The goats were housed in individual pens (1.5 m^2^) which were equipped with drinker and feeding trough. Water was provided *ad libitum*. Feed was supplied with daily adjustments to allow around 10% orts.

The experimental diets consisted of the inclusion rates of PKC: 0, 80, 160 and 240 g kg^−1^ of dry matter (DM) [12]. The PKC was obtained from “LUZIPALMA extração de óleos vegetais Ltd.a” (Rodovia Aratuipe, 45400000, Valença, BA/Brazil). The diets were supplied as a total mixture ration, twice daily (08:00 h and 15:00 h). A forage:concentrate ratio of 40:60 was adopted, with maize silage used as the forage. Diets were formulated according to the NRC [13] to meet the requirements for maintenance and milk production of lactating goats.

Hand milking was performed at 07:00 h, after pre-dipping with a 0.5% glycerin iodine solution. After milking, post-dipping was performed using a 0.5% glycerin iodine solution. Hygiene procedures were followed to avoid mammary gland infections. The average body weight of the goats was obtained after the pre-adaptation period and on the first and last day of each experimental period (Day 1 and 14). The weighing were made before supplying the diets, in the morning and with the help of an electronic scale.

### 2.3. Intake and Apparent Digestibility of Nutritional Components

Intake evaluation was performed between days 12 and 14 of each period collecting supplied diets and orts. Intake was calculated as the difference between the amount of the component present in the feed supplied and in the orts.

The apparent digestibility was determined by the indirect method (spot collection), collecting the feces directly from the rectal ampulla. The collections were made in different hours on three consecutive days: 12th (08:00 h and 14:00 h), 13th (10:00 h and 16:00 h) and 14th (12:00 h and 18:00 h) experimental days [14]. The feces samples were subjected to pre-drying in a forced ventilation oven at 55 °C; ground in a mill with a 1 mm sieve and mixed in equal proportions to form a composed sample.

Fecal excretion was estimated using non-digestible neutral detergent fiber (NDFi) as an internal marker [15]. Apparent digestibility was calculated according to Berchielli et al. [16].

### 2.4. Feeding Behavior

The feeding behavior was analyzed for 24 h, in intervals of 5 min. The observations were made on the 11th day of each experimental period. Feeding, rumination and idleness activities were recorded according to the methodology proposed by Johnson and Combs [17].

These data were recorded by 8 trained evaluators, distributed in pairs during time intervals of 2 h. During these 2 h, the two evaluators performed feeding behavior every 5 min; after this time, two new evaluators performed the same exercise for the same amount of time. The evaluators were positioned to minimally interfere with the feeding behavior of the animals. During the night evaluations, the environment was maintained with artificial lighting.

The feeding, rumination and idleness episodes were obtained by the number of periods of time that animals performed each activity. Feed and rumination efficiency results for DM and neutral detergent fiber (NDF) were calculated by dividing the intake of these nutrients by the time spent feeding and ruminating, respectively. The data referring to the feeding behavior were obtained according to the methodologies described by Bürger et al. [18].

### 2.5. Chemical Analysis

During the experimental period, samples of ingredients and orts were collected and dried in a forced-air oven at 55 °C for 72 h. Once dried, the samples were divided into two portions that were ground in a Wiley knife mill to 1-mm particles for chemical composition analysis; or 2-mm particles. These samples were then used to measure the DM (934.01), ash (930.05), crude protein (CP, 981.10), and ether extract (EE, 920.39) contents following the methodology proposed by the Association of Official Agricultural Chemists (AOAC) [19].

Acid detergent fiber (ADF) and NDF were determined as proposed by Van Soest et al. [20] with the adaptations described by Mertens [21]. NDF corrections for ash and protein (NDFap) were performed according to Sniffen et al. [22] and Licitra et al. [23], respectively. Lignin was determined according to the AOAC method 973.18 [19], by immersing the ADF residue in a 72% sulfuric acid solution.

Indigestible neutral detergent fiber (iNDF) was determined by in situ incubation of samples inside non-woven fabric (TNT) bags weighing 100 g m^−2^, following the methodology described by Valente et al. [24]. Potentially digestible neutral detergent fiber (pdNDF) was determined as the difference between neutral detergent fiber corrected for ash and protein (NDFap) and iNDF.

To estimate non-fibrous carbohydrates (NFC) and total digestible nutrients (TDN) were used the equations proposed by Hall [25] and Da Cruz et al. [26], respectively.

### 2.6. Production, Composition, and Quality of the Milk

Milk production was determined per animal and per day during the last four days of each experimental period. Fat-corrected milk production (FCMP 4%) was obtained using the formula described in the NRC [27]: FCMP 4% = 0.4 × milk production (g) + 15 × milk fat (g).

Milk samples were collected and placed in plastic bottles containing the preservative 2-bromo-2-nitropropane-1,3-diol (bromopol) for analysis of protein, fat, lactose, urea nitrogen and total solids, using the Bentley-2000 infrared analyzer; and somatic cell count, using the Somacount-500 instrument. These analyses were performed at the laboratory of Clínica do Leite at ESALQ/USP, in Piracicaba-SP, Brazil.

Milk production (MPE) and fat-corrected milk production (FCMPE) efficiencies were obtained as follows:

MPE = milk production (g)/dry matter intake (g).

FCMPE = fat-corrected milk production (g)/dry matter intake (g).

### 2.7. Blood Metabolites

Blood samples were collected by puncture of the jugular vein on day 14 of each experimental period. Collections were made 4 h after the first feeding, using vacuum tubes (vacutainer). Blood samples were then centrifuged at 3500 rpm for 15 min to obtain serum, which was stored in identified eppendorfs and stored in a −20 °C freezer for further analysis.

The colorimetric method and commercial Doles kits (Doles Reagentes Ltd., Goiânia, Goiás, Brazil) were used to determine the serum concentrations of albumin, total protein and urea. Readings were made in spectrophotometer (AJX-1900, Micronal S.A., São Paulo, Brazil).

### 2.8. Nitrogen Balance

Spot urine samples were collected on day 13 of each experimental period, approximately 4 h after the first feeding. After collection, aliquots of 10 mL of urine were diluted in 40 mL of 0.036N sulfuric acid, as described by Valadares et al. [28]. Immediately, samples were placed in labeled plastic containers and frozen for further analysis.

The creatinine content of the samples was determined using a commercial kit (Labtest^®^, Lagoa Santa, Minas Gerais, Brazil) and a spectrophotometer (AJX-1900, Micronal SA, São Paulo, Brazil). This value was used to estimate the daily urinary excretion following the formula proposed by Fonseca et al. [29], which considers an average creatinine excretion for lactating goats of 26.05 mg kg^−1^ of body weight (BW).

Daily urinary excretion (L day^−1^) = (26.05 × BW (kg))/creatinine concentration of the sample (mg L^−1^).

The nitrogen balance was obtained according to Zeoula et al. [30].

### 2.9. Statistical Analysis

For the analyses, SAS statistical software version 9.2 (Statistical Analysis System, 2009) [31] was used. The variables of intake, digestibility, feeding behavior, milk production and nitrogen metabolism were assessed according to a triplicated 4 × 4 Latin Square. The mathematical model below was applied:Ŷ_ijkl_ = μ + LS_i_ + A(LS_i_)_j_ + P_k_ + PKC_l_ + LS_i_ × PKC_l_ + Ɛ_ijkl_;
where Ŷ_ijkl_ = dependent variable; μ = overall mean; LS_i_ = fixed effect of the Latin Square (i = 1, 2 and 3); A(LS_i_)_j_ = random effect of the animal into the Latin Square (j = 1, 2, 3 and 4); P_k_ = random effect of the period (k = 1, 2, 3 and 4); PKC_l_ = effect of the PKC inclusion rate (l = 0, 80, 160 and 240 g kg^−1^); LS_i_ × PKC_l_ = fixed effect of the interaction between Latin Square and PKC inclusion rate; and Ɛ_ijkl_ = random experimental error associated with each observation, with NID ~ (0, σ2) assumption. 

Furthermore, the effect of the PKC inclusion rate was evaluated using Orthogonal Polynomial Contrasts to determine the linear (−3, −1, +1, +3) and quadratic (+1, −1, −1, +1) effects. For all the evaluations, the level of 5% probability of type I error (*p* ≤ 0.05) was considered.

No interaction between treatment and racial group was observed for any of the variables studied.

## 3. Results

### 3.1. Intake and Apparent Digestibility

There was a linear reduction (*p* < 0.05) in the intake of nutritional components due to the PKC inclusion in the diet, except the EE intake (*p* = 0.015), which was influenced quadratically (Table 1).

There was no difference in the apparent digestibility of CP (*p* > 0.05), NDF (*p* > 0.05), EE (*p* > 0.05) and NFC (*p* > 0.05) with the PKC inclusion. However, the apparent digestibility of DM (*p* = 0.031) and TDN (*p* = 0.026) were linearly reduced (Table 1).

### 3.2. Feeding Behavior

There was a linear reduction in the feeding and rumination efficiencies of DM and NDF (*p* < 0.01). The other parameters of feeding behavior were not influenced by the inclusion of increasing PKC levels in the diet (Table 2).

### 3.3. Milk Production and Composition

Milk production (*p* < 0.001) and defatted dry extract content (*p* < 0.001) decreased linearly. The PKC inclusion did not influence the ureic nitrogen in milk and milk production efficiency (*p* > 0.05) (Table 3), with mean values of 26.3 mg dL^−1^ and 0.52 L of milk produced per kg of DM consumed, respectively.

### 3.4. Blood Metabolites

The inclusion of PKC reduced the content of serum urea (*p* = 0.001) (Table 4). Serum concentrations of albumin, total proteins, globulin and the albumin:globulin ratio were not influenced (*p* > 0.05) by the PKC inclusion in the diet (Table 4), with mean values of 1.56 g dL^−1^, 4,98 g dL^−1^, 3.40 g dL^−1^ and 0.5, respectively.

### 3.5. Nitrogen Balance

The inclusion of increasing levels of PKC promoted a gradual reduction of the nitrogen intake (*p* = 0.001), nitrogen excretion in feces (*p* = 0.001), nitrogen excretion in milk (*p* = 0.001), nitrogen retained (*p* = 0.001) and nitrogen digested (*p* = 0.001) (Table 5). The nitrogen excretion in urine was not altered with the PKC inclusion rates, with a mean value of 18.9 g d^−1^.

The regression equation of all the parameters that showed significant values of P, are described in Table 6.

## 4. Discussion

### 4.1. Intake and Apparent Digestibility

According to Hoffman [32], goats are classified as intermediate consumers and are the most selective domestic ruminants. Therefore, the observed reduction in DM intake (DMI) and in all nutritional components is associated with low acceptability of PKC by lactating goats. Large amounts of PKC were observed in the refusals of this study. The predominance of this by-product in the refusals was also noted by Silva et al. [7]. These authors evaluated high-concentrate diets for culling goats and observed a higher amount of PKC in the refusal from the inclusion rate of 240 g kg^−1^ DM.

There was a reduction in the CP intake by 55.51% and NFC intake by 70.62% with the inclusion rate of 240 g kg^−1^ DM in the diet, compared to the diet without the inclusion of PKC. Therefore, the reduction of the DM digestibility may have occurred due to the decrease in the amount of rapidly fermentable substrate, which is necessary for microbial growth. It is important to note that the rate of microbial growth is one of the factors that interfere with digestibility [33].

Furthermore, the relationship between the intake of NFC and NDF (NDF/NDF) was modified with the inclusion of PKC in the diet, going from 1.16 to 0.64 from the diet without PKC to the diet with 240 g kg^−1^ DM. This confirms the hypothesis of a reduced digestibility of DM due to the lower availability of fermentable substrate in the rumen for microbial growth [32]. A similar result was observed by Song et al. [34] who tested the reduction of the NFC/NDF ratio (1.16 and 1.66) and observed a lower efficiency of microbial protein synthesis for animals that consumed a lower NFC/NDF ratio. The reduction of the DM digestibility was also observed by Silva et al. [7], who studied the inclusion of PKC in high-concentrate diets for culling goats.

### 4.2. Feeding Behavior

The time spent on feeding did not change with the inclusion of PKC in the diet of lactating goats. It is possible to affirm that the effect observed in the feeding time of the goats is the result of avoiding the concentrate intake. This behavior indicates that when PKC was included in the diet, the goats spent time practicing diet selection rather than eating the feed. Therefore, although the DM intake decreased with the inclusion of PKC, the feeding time did not decrease due to the effect promoted by selective behavior. This selective behavior avoiding the intake of PKC was also observed when PKC was included in high-concentrate diets for feedlot goats [35].

Goats fed the PKC diets predominantly consumed the forage of the diet. Corn silage has larger particle size compared to concentrate particle size and this effect can change rumination behavior. This theory is confirmed by Schultz et al. [36], who studied NDF levels and different particle sizes in lactating goat diets and reported that increasing particle size causes an increasing linear effect on feeding time, with an increase of 4 min/day per each cm increase in particle size.

Similar to feeding time behavior, PKC inclusion negatively affected DM and NDF intake but was not significant in affecting rumination time. This behavior was not affected by the effect promoted by the selective behavior and the consumed particle size. The increase in rumination time as a function of the increase in particle size was also reported by Arowolo et al. [37], who observed that, by increasing the forage particle size from 20 to 100 mm in goat diets, there was an increase of 20% in feeding time and 16% in rumination time.

Feeding and rumination efficiencies are estimated parameters based on the intake of the nutritional component (DM or NDF) and the time spent on the respective activity. In the current study, the times spent on feeding and ruminating were not affected by the inclusion of PKC in the diet of lactating goats. However, DM and NDF intake was reduced; thus explaining why feeding and rumination efficiencies were also reduced. This behavior can be corroborated with that found by Silva et al. [7] who also observed a reduction in feeding and rumination efficiencies in culling goats fed diets containing PKC.

### 4.3. Milk Production and Composition

The observed reduction in milk production of goats fed diets with increasing levels of PKC can be explained by decreased intake of nutritional components. According to the NRC [13], the nutritional requirement of CP and TDN of a lactating goat with an average production of two kg of milk per day is 229.9 g/day and 1219.4 g/day, respectively. The values obtained by means of the regression equations indicate that the CP intake for the production needs were met only by diets with PKC inclusion of up to 80 g kg^−1^ DM. On the other hand, the NDT intake requirements were only met with the diet without PKC inclusion.

When the NFC and NDF intake ratio is modified, there is also a change in the ruminal microbiota and, consequently, in the proportion of ruminal fermentation products. In this sense, the reduction of NFC intake leads to a lower production of propionic acid [38]. According to Kennedy et al. [39], propionate can contribute 61 to 95% of glucose synthesis and approximately 60% of glucose is used for lactose production in the mammary gland [40]. It is important to highlight that most of the glucose produced from propionate is used for the production of lactose, which is the main responsible for the increase in water in milk due to its hygroscopic property [41,42,43,44]. Therefore, the reduction in the intake of NFC and TDN are the main responsible for the decrease in milk production of goats subjected to diets with increasing levels of PKC.

Our findings are similar to those of Santos et al. [44] who carried out a meta-analysis study to demonstrate the effects of the use of PKC in diets for lactating cows. These authors observed that the reduction in NFC intake due to the inclusion of PKC promoted negative effects on intake and ruminal digestibility, promoting a reduction in energy intake for milk production.

Milk production efficiency consists of the relationship between milk production and DM intake. It was observed that the reduction in DM intake was proportionally reflected in the decrease in milk production, which indicates that the nutrient supply was not sufficient to meet the productive requirement of lactating goats fed PKC diets. Although milk production efficiency was not affected by the inclusion of PKC in the diet of lactating goats, it is possible to state that milk production was dependent on DM intake.

### 4.4. Blood Metabolites

Dietary nitrogen can be used by ruminal microorganisms in the form of amino acids or ammonia [16]. In this sense, changes in CP intake are reflected in the ammonia content in the rumen [45]. The nitrogen that is degraded in the rumen and is not used for the production of microbial protein is absorbed in the rumen wall in the form of ammonia and transported to the liver. In this organ, ammonia is transformed into urea, a less toxic compound for the organism, which returns to the rumen through the blood in an efficient recycling process [40,46]. In this sense, the reduction of 0.47 mg dL^−1^ of serum urea per g of PKC included in the diet is associated with a lower CP intake due to the inclusion of the by-product. Although a decrease in serum urea was observed, it is within the reference values for goats, which is between 20 and 60 mg dL^−1^ [47].

### 4.5. Nitrogen Balance

The low acceptability of PKC, a factor that influenced the reduction of DM intake, also promoted lower nitrogen intake. Nitrogen present in feces is generally the indigestible nitrogen of the diet. Although the inclusion of PKC promoted a greater amount of iNDF in the diets; the reduction in the intake of the nutritional components promoted by the by-product resulted in the reduction of the nitrogen present in the feces. Corroborating our study, Silva et al. [7] also observed a reduction in fecal nitrogen excretion in culling goats fed high-concentrated diets with increasing inclusion rates of PKC in the diet.

For the synthesis of nitrogenous compounds in milk to occur, the mammary gland must have amino acids from microbial protein and from the diet [48]. For this, it is necessary to have a substrate for the production of microbial protein that involves the nutritional components of the diet, necessary to obtain sufficient energy for protein synthesis. Therefore, the reduction in nutrient intake possibly limited the supply of nutritional sources for microbial synthesis and, consequently, for the mammary gland, which is associated with the reduction of nitrogen excretion in milk.

Despite the reduction in retained nitrogen as a function of PKC, only the inclusion of 240 g kg^−1^ of PKC had a negative value, indicating that, at this inclusion rate, the animal needed to mobilize amino acids from body tissues to maintain it nutritional requirements for maintenance and milk production [16].

The current prices of the main ingredients used in ruminant feed, soybean meal and corn have increased in recent years. The PKC price represents 37.6% and 19.6% of the price of these ingredients, respectively [49]. In this sense, it is important to highlight that reducing the price of the diet without affecting animal performance is one of the main objectives of rural and industrial producers.

Our findings are important for future research. We can promote considering in future research work the evaluation of PKC levels in refusals, promote research on the use of other ingredients (e.g., cane molasses, salt) to improve the palatability of PKC and add the economic evaluation of the inclusion of PKC in the diet. Nutritionally, PKC has great potential to be included in ruminant diets. With further studies, it is likely that the use of this ingredient could be increased to levels to compete with corn or soybean meal. This increase in use would promote greater production of this by-product, resulting in improvements in the socioeconomic levels of the population.

## 5. Conclusions

The use of PKC in diets for lactating goats reduced nutrient intake and milk production. In this sense, the moderate use of up to 80 g kg^−1^ DM is recommended because it is an ingredient of lower commercial value and that at this level it does not significantly affect milk production. Levels higher than 80 g kg^−1^ DM promoted a pronounced rejection of the diet due to the low acceptance of this by-product by lactating goats in feedlots.

## Figures and Tables

**Table 1 animals-12-02323-t001:** Intake and apparent digestibility of nutritional components in lactating goats fed diets with increasing levels of palm kernel cake.

Variable	Palm Kernel Cake (g kg^−1^)				SEM	*p*-Value	
	0	80	160	240		Linear	Quadratic
**Final body weight (kg)**	45.85	46.82	46.76	44.90	1.42	0.208	0.009
**Intake (g day^−1^)**							
Dry matter	1677.9	1682.6	1251.0	842.1	78.33	<0.001	0.618
Organic matter	1597.4	1584.1	1170.1	782.5	75.86	<0.001	0.618
Crude protein	316.2	268.9	200.9	130.6	13.57	<0.001	0.430
Ether extract	102.3	107.8	88.76	67.2	4.49	<0.001	0.015
Neutral detergent fiber	577.2	576.7	423.0	247.6	25.92	<0.001	0.216
Potentially digestible neutral detergent fiber	281.3	255.9	193.0	62.2	16.00	<0.001	0.225
Non-fiber carbohydrates	633.8	546.3	332.3	207.8	35.38	<0.001	0.523
Total digestible nutrients	1390.0	1245.3	884.3	619.5	62.19	<0.001	0.491
**Apparent digestibility (g Kg DM^−1^)**							
Dry matter	675.0	652.0	646.0	634.0	5.20	0.031	0.290
Organic matter	645.8	645.8	647.4	633.6	5.89	0.528	0.577
Crude protein	686.0	672.0	681.0	629.0	4.17	0.771	0.459
Ether extract	920.0	929.0	927.0	894.0	3.86	0.314	0.734
Neutral detergent fiber	400.0	361.0	430.0	462.0	9.69	0.113	0.131
Non-fiber carbohydrates	876.0	921.0	889.0	906.0	6.44	0.150	0.224
Total digestible nutrients	771.0	742.0	748.0	735.0	7.30	0.026	0.407

**Table 2 animals-12-02323-t002:** Feeding Behavior of lactating goats fed diets with increasing levels of palm kernel cake.

Variable	Palm Kernel Cake (g kg^−1^)				SEM	*p*-Value	
	0	80	160	240		Linear	Quadratic
**Time per activity (min day^−1^)**							
Feeding	242	258	304	255	9.14	0.284	0.061
Rumination	394	394	376	371	9.50	0.282	0.874
Idling	803	789	761	815	14.18	0.938	0.183
**Feeding efficiency (g h^−1^)**							
Dry matter	476.3	404.7	250.3	229.8	23.86	<0.001	0.358
Neutral detergent fiber	160.3	152.8	105.3	107.0	8.33	0.001	0.636
**Rumination efficiency (g h^−1^)**							
Dry matter	290.5	250.4	202.1	153.1	12.22	<0.001	0.680
Neutral detergent fiber	97.2	94.3	84.7	71.0	4.12	0.001	0.229
**Periods per activity (N° of episodes day^−1^)**							
Feeding	16.0	15.0	16.0	13.0	0.59	0.208	0.409
Rumination	26.0	26.0	25.0	26.0	0.60	0.658	0.448
Idling	37.0	36.0	34.0	34.0	0.72	0.099	0.681

**Table 3 animals-12-02323-t003:** Milk production and composition of lactating goats fed diets with increasing levels of palm kernel cake.

Variable	Palm Kernel Cake (g kg^−1^)				SEM	*p*-Value	
	0	80	160	240		Linear	Quadratic
**Performance (g day^−1^)**							
Milk production	933.5	862.2	763.8	532.5	69.70	<0.001	0.087
Fat-corrected milk production	906.4	946.0	774.5	509.4	63.40	<0.001	0.004
**Milk composition (%)**							
Defatted dry extract	8.99	9.14	8.70	8.44	0.10	0.001	0.152
Ureic nitrogen (mg dL^−1^)	26.4	26.9	26.2	25.9	0.43	0.229	0.750
**Milk production efficiency (L kg^−1^DMI)**							
Milk production	0.48	0.51	0.53	0.54	0.08	0.125	0.696
Fat-corrected milk production	0.47	0.57	0.53	0.54	0.06	0.266	0.225

**Table 4 animals-12-02323-t004:** Blood metabolites of lactating goats fed diets with increasing levels of palm kernel cake.

Variable	Palm Kernel Cake (g kg^−1^)				SEM	*p*-Value	
	0	80	160	240		Linear	Quadratic
Albumin (g dL^−1^)	1.51	1.61	1.61	1.49	0.07	0.946	0.137
Total proteins (g dL^−1^)	5.09	4.58	4.73	5.50	0.21	0.460	0.175
Globulin (g dL^−1^)	3.56	2.89	3.12	4.02	0.22	0.510	0.098
Albumin:globulin ratio	0.46	0.49	0.49	0.44	0.23	0.352	0.151
Urea (mg dL^−1^)^1^	59.3	64.2	52.6	52.1	1.62	<0.001	0.320

**Table 5 animals-12-02323-t005:** Nitrogen balance in lactating goats fed diets with increasing levels of palm kernel cake.

Variable	Palm Kernel Cake (g kg^−1^)				SEM	*p*-Value	
	0	80	160	240		Linear	Quadratic
**Nitrogen (g d^−1^)**							
Ingested	50.6	43.2	32.1	20.9	2.22	0.001	<0.430
Excreted in feces	10.5	9.51	6.46	4.58	0.51	0.001	0.505
Excreted in milk	5.02	4.73	3.84	2.62	0.35	0.001	0.169
Excreted in urine	16.3	21.1	20.3	17.7	1.59	0.727	0.103
Retained	18.8	7.89	1.54	−4.02	2.16	0.001	0.431
Digested	40.1	33.7	25.6	16.3	1.78	0.001	0.453

**Table 6 animals-12-02323-t006:** Regression equations for the variables performed in lactating goats fed diets with increasing levels of palm kernel cake.

Variable (Y)	Type	Equation	Determination Coefficient (R^2^)
**Final body weight (kg)**	Quadratic	Y = 45.81 + 0.02292PKC + 0.00011PKC^2^	0.82
**Intake (g day^−1^)**			
Dry matter	Linear	Y = 1919.29 − 4.1961PKC	0.99
Organic matter	Linear	Y = 1852.28 − 4.0757PKC	0.99
Crude protein	Linear	Y = 322.86 − 0.7809PKC	0.99
Ether extract	Quadratic	Y = 103.02 + 0.08788PKC − 0.00102PKC²	0.82
Neutral detergent fiber	Linear	Y = 659.94 − 0.9367PKC	0.94
NDFpd ^1^	Linear	Y = 399.56 − 0.5442PKC	0.93
Non-fiber carbohydrates	Linear	Y = 764.61 − 2.2499PKC	0.99
Total digestible nutrients	Linear	Y = 1431.45 − 3.30PKC	0.98
**Apparent digestibility (g Kg DM^−1^)**			
Dry matter	Linear	Y = 669.1 − 0.133PKC	0.79
Total digestible nutrients	Linear	Y = 76.53 − 0.1317PKC	0.73
**Feeding efficiency (g h^−1^)**			
Dry matter	Linear	Y = 475.25 − 1.1157PKC	0.92
Neutral detergent fiber	Linear	Y = 162.27 − 0.2555PKC	0.81
**Rumination efficiency (g h^−1^)**			
Dry matter	Linear	Y = 293.34 − 0.5809PKC	0.99
Neutral detergent fiber	Linear	Y = 100.14 − 0.1121PKC	0.92
**Performance (g day^−1^)**			
Milk production	Linear	Y = 969.20 − 1.6263PKC	0.92
Fat-corrected milk production	Quadratic	Y = 912.31 + 1.1524PKC − 0.01190PKC^2^	0.80
**Milk composition (%)**			
Defatted dry extract	Linear	Y = 9.1745 − 0.00285PKC	0.80
**Blood Urea (mg dL^−1^)**	Linear	Y = 63.06 − 0.4710PKC	0.63
**Nitrogen (g d^−1^)**			
Ingested	Linear	Y = 51.6571 − 0.1249PKC	0.99
Excreted in feces	Linear	Y = 10.87 − 0.0259PKC	0.97
Excreted in milk	Linear	Y = 5.456 − 0.0111PKC	0.97
Retained	Linear	Y = 16.8965 − 0.09321PKC	0.98
Digested	Linear	Y = 40.79 − 0.09902PKC	0.99

^1^ NDFpd, potentially digestible neutral detergent fiber.

## Data Availability

The study did not report any data.

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
