# Peer review of "Palm Kernel Cake in Diets for Lactating Goats: Intake, Digestibility, Feeding Behavior, Milk Production, and Nitrogen Metabolism"

_animals, 2022, doi:10.3390/ani12182323_

Round 1

Reviewer 1 Report

In the introduction, please insert the novelty in this study, palm kernel cake is already used for ruminants as a feed resource, and please try to emphasize that with the parameters that were used in this study (e.g., nutrient digestibility, milk production, ...). Additionally, the authors hypothesized that goats have specific characteristics and feeding behaviors that can influence the intake of different feed sources, yes it is true for goats but you introduced the diet as a TMR, so the selectivity by the animal may not be an issue here, so please change the hypothesis, may you can something about the secondary metabolites that may found in the PKC (e.g., mannan) or the high fiber content compared to other meals (e.g., soybean) or grains (e.g., corn) that may affect the digestibility and feed intake of the animals.

Another important issue in the introduction is that you have to refer to which part of the diet you will replace by the experimental PKC and why.

another limitation of this study is the experimental time used for sample collections, especially the milk samples.   

Line 24, recommended for what? please specify.

Line 70 105±5 days in milk!! which stage of milk yield? for these goat breeds how many days for the lactation period? is this period is sufficient to do a 4×4 Latin square design?

Line 83, for which animal?

line 84, how many days the milk was collected?

Table 1 as the same as what was published by Ferreira et al., 2021 (Animals 202111(12), 3501; https://doi.org/10.3390/ani11123501), even it is your work but you have to refer to the published article and not double the same table in two different articles

The source of PKC has to be interesting, commercially available from....In the statics, please insert the significant experimental value (e.g., P<0.05)...

Are results of intake and milk composition obtained from your previous work (Ferreira et al., 2021)? or it is a different experiment? if it is a new experiment why do you repeat the experimental parameters?

Line 251, but the diet was TMR!!

You don't measure the ruminal propionate in this study, so the discussion for propionate might be invalid.

Line 333, you used just one ratio between forage and concentrate in the experimental diet. 

Author Response

Comments to the author

Correction and answers

Reviewer - 1

General comments

In the introduction, please insert the novelty in this study, palm kernel cake is already used for ruminants as a feed resource, and please try to emphasize that with the parameters that were used in this study (e.g., nutrient digestibility, milk production, ...). Additionally, the authors hypothesized that goats have specific characteristics and feeding behaviors that can influence the intake of different feed sources, yes it is true for goats but you introduced the diet as a TMR, so the selectivity by the animal may not be an issue here, so please change the hypothesis, may you can something about the secondary metabolites that may found in the PKC (e.g., mannan) or the high fiber content compared to other meals (e.g., soybean) or grains (e.g., corn) that may affect the digestibility and feed intake of the animals.

Dear reviewer, thank you very much for your time, attention and contributions to improve our manuscript.

Dear reviewer, we believe that the novelty of our study is the use of PKC in diets for goats. However, to improve this answer, we have added to the text that our results are needed to augment the limited information on the use of PKC in goat diets.

The parameters used in the current experiment were emphasized in the introduction.

Dear reviewer, the diet was mixed (TMR) a few minutes before feeding the animals (it was not pelleted; just a simple mix between concentrate and silage). Although the silage is wet forage and theoretically it would mix very well with the concentrate; we see in the refusals large amounts of the characteristic PKC particles.

As suggested by you, we think of antinutritional compounds, toxic compounds, fiber content, among others that make up the PKC as responsible for our results. If not, the PKC's notoriety in the refusals is something we couldn't help. Unfortunately, we lost the photographs of these refusals when our camera was stolen from our laboratory facilities.

Another important issue in the introduction is that you have to refer to which part of the diet you will replace by the experimental PKC and why.

Thank you very much for your suggestion. Dear Reviewer: We use PKC as a dietary inclusion. Coincidentally, when PKC was added, two ingredients were replaced. However, our goal was to include PKC and not replace other specific ingredients. In addition, we believe that the use of the word “replace” could promote different points of view about the ingredients that these diets contain. For these reasons, we would like to ask you to allow us to keep the word "inclusion" in the current manuscript and not talk about replacing specific ingredients. We think that the objective of including by-products is to reduce costs.

Another limitation of this study is the experimental time used for sample collections, especially the milk samples.

Thank you very much for your suggestion.

Dear reviewer, we understand your concern. However, according to statistical models, the days of feeding of the animals are more important to find significant results of the effect of the diets. The days of adaptation are in accordance with the following studies:

Machado, M.G., et al. (2016). Evaluation of the length of adaptation period for changeover and crossover nutritional experiments with cattle fed tropical forage-based diets. Animal Feed Science and Technology, 222, 132-148. https://doi.org/10.1016/j.anifeedsci.2016.10.009.

Price, T.P., et al. (2021). Short-term adaptation of dairy cattle production parameters to individualized changes in dietary top dress. Animals, 11(12), 3518. https://doi.org/10.3390/ani11123518.

Specific comments

Line 24, recommended for what? please specify.

Thank you very much for your observation. We add information in the text being specific.

Line 70 105±5 days in milk!! which stage of milk yield? for these goat breeds how many days for the lactation period? is this period is sufficient to do a 4×4 Latin square design?

Thank you very much for your observation.  Both breeds have an average lactation period of 210 days, so 105 days correspond to the middle of the lactation period.

When we work with the evaluation of feeds for lactating animals, the concern is that it is not at the beginning of lactation due to the negative energy balance (NEB), in this way, we believe that the lactation phase is adequate for the experimental design.

Line 83, for which animal?

Thank you very much for your observation. We add information in the text being specific.

Line 84, how many days the milk was collected?

Thank you very much for your observation. The milk was collected the last four days of each experimental period. This information is found in the subtopic 2.6.

Table 1 as the same as what was published by Ferreira et al., 2021 (Animals 2021, 11(12), 3501; https://doi.org/10.3390/ani11123501), even it is your work but you have to refer to the published article and not double the same table in two different articles.

Thank you very much for your suggestion. The table was removed and the previous published manuscript was cited.

The source of PKC has to be interesting, commercially available from....In the statics, please insert the significant experimental value (e.g., P<0.05)…

Thank you very much for your suggestion. The information was added.

Are results of intake and milk composition obtained from your previous work (Ferreira et al., 2021)? or it is a different experiment? if it is a new experiment why do you repeat the experimental parameters?

Thank you very much for your observation. Dear reviewer, the data is the same; however, the units (kg, %) are different. In addition, in data processing we use a more stringent statistical analysis with a lower margin of outliers.

It is important to highlight that these parameters help us a lot to explain or correlate the effect of PKC with the other variables found in our results. For this reason, we ask that you let us maintain these variables. However, if you want us to remove it, we will remove it.

Line 251, but the diet was TMR!!

Thank you very much for your observation. Dear reviewer, we understand that it is hard to believe. How do goats separate dry ingredients from wet silage? We don't know, but the refusals contained quite a bit of PKC. This behavior was also observed in another experiment developed by our study group (manuscript cited Silva et al., 2021).

You don't measure the ruminal propionate in this study, so the discussion for propionate might be invalid.

Thank you very much for your observation. Dear reviewer, it is true. We did not evaluate ruminal propionate; however, this is only a hypothesis that attempts to explain our results. We would like to keep this text; however, if you consider that this text is not valid as a hypothesis, we will eliminate it as suggested.

Line 333, you used just one ratio between forage and concentrate in the experimental diet.

Thank you very much for your observation. Dear reviewer, it is true. We use just one ratio between forage and concentrate. However, this text discusses changes in the ratio of CNF and NDF intake (as opposed to the ratio of these components in the formulated diet) and how this change might affect rumen metabolism.

Reviewer 2 Report

In the present work, the authors studied the use of PKC (at levels of 80, 160 and 240 g kg-1) in goat diets. The findings are very promising, and authors suggested the use of PKC up to 80 g kg-1 for the diet of goats without any noteworthy drawback in comparison to the conventional diet. In general, the paper is well-written, even though English editing in some parts is needed. The aim of the work is clear, and the results are well-presented. However, I found some crucial flaws that prevent a positive suggestion for paper’s publication in its current form.

1)     The number of animals used for this study is too low (12), making the experimental design very weak and thus, the findings cannot strongly be supported.

2)     The novelty of the present work is a bit low. I suggest authors to revise the scope of their work and highlight in a better way the novelty

3)     For my point of view, a final paragraph in the discussion section, highlighting the main findings of this work, as well as how the industry and stakeholders may be benefited by these findings, is recommended.

4)     Further, authors should propose/highlight how this work could contribute to the “next research steps” and which are future prospects.

5)     The English language needs to be improved in the whole manuscript

Author Response

Comments to the author

Correction and answers

Reviewer-2

General comments

In the present work, the authors studied the use of PKC (at levels of 80, 160 and 240 g kg-1) in goat diets. The findings are very promising, and authors suggested the use of PKC up to 80 g kg-1 for the diet of goats without any noteworthy drawback in comparison to the conventional diet. In general, the paper is well-written, even though English editing in some parts is needed. The aim of the work is clear, and the results are well-presented. However, I found some crucial flaws that prevent a positive suggestion for paper’s publication in its current form.

Dear reviewer, thank you very much for your time, attention and contributions to improve our manuscript.

1) The number of animals used for this study is too low (12), making the experimental design very weak and thus, the findings cannot strongly be supported.

Thank you very much for your observation. Dear reviewer, we understand your concern; however, the experimental design is a triplicate Latin Square. A Completely Randomized Design using 4 treatments and 10 animals per treatment adds a total of 39 degrees of freedom. This type of study is widely used and accepted.

The degrees of freedom in this study are 47 considering a triplicate Latin square with 4 treatments and 4 animals per treatment.

2) The novelty of the present work is a bit low. I suggest authors to revise the scope of their work and highlight in a better way the novelty

Thank you very much for your suggestion. Dear reviewer, we believe that the novelty of our study is the use and ideal level of PKC found in dairy goat diets. In accordance with the following suggestion, we add the importance of price in the last paragraph of the discussion as a highlight.

3) For my point of view, a final paragraph in the discussion section, highlighting the main findings of this work, as well as how the industry and stakeholders may be benefited by these findings, is recommended.

Thank you very much for your suggestion. Dear reviewer, we add the importance of price in the last paragraph of the discussion as a highlight.

4) Further, authors should propose/highlight how this work could contribute to the “next research steps” and which are future prospects.

Thank you very much for your suggestion. Dear reviewer, we add the suggestion in the last paragraph of the discussion as a highlight.

5) The English language needs to be improved in the whole manuscript

Thank you very much for your suggestion. Dear reviewer, we sent the manuscript to a native English speaker. In addition, the Journal has assistance before publication.

Reviewer 3 Report

Article ID animals 1882155

Palm Kernel Cake in Diets for Lactating Goats: Intake, Digestibility, Feeding Behavior, Milk Production, and Metabolic Profile

GENERAL REMARKS

Dear authors,

I have evaluated your manuscript (ID animals 1882155) concerning the effects of including palm kernel cake in the diet of dairy goats. I believe the manuscript is fascinating since it provides useful information for the "safe" reuse of palm kernel cake in dairy goats' diets, for which little information is available as far as I know. Therefore, the research's contribution to the literature could be greatly appreciated. Nevertheless, in my opinion, the manuscript deserves to be revised in different parts. In addition, bibliographic references have been omitted, making the revision work very difficult and, at least partially, incomplete. So, I suggested a major revision. My suggestions are detailed below, section by section. Hoping to have contributed to improving the manuscript quality. Good works.

SPECIFIC COMMENTS

L 3: I believe that the title excludes an important part of the research (the N balance) while it provides a partially true indication since the metabolic profile (generally referred at blood) concerns exclusively some blood markers of the N cycle. To point out that the metabolic profile has been studied only regarding N metabolism and that, at the same time, the N balance between treatments has been measured, I suggest replacing "Metabolic profile" (considered too generic in this case) with "Nitrogen Metabolism". This generic term, in my opinion, can be fruitfully used to summarize both the effects on plasma metabolites and the N balance as a whole. Thanks.

L 13-14: I suggest avoiding the adjective "ideal" (too pompous), which can be replaced by "optimal inclusion rate" that fits better the study purposes, in my opinion. Furthermore, it would be advisable to specify in which farming system the palm kernel cake was used or, more simply, in which type of basal diet it's incorporated. In my opinion, the emphasis on this detail is important, as the trial was performed on "feedlot" goats fed with a silage-based diet, which results in both unusual conditions for goat farming in large parts of the world. Thanks.

L 15 (and throughout the manuscript): see the comment regarding line 3 about the replacement of metabolic profile with nitrogen metabolism. Thanks.

L 16: I apologize for the pedantry, but information such as body weight and days in milk is probably too superfluous in this part of the manuscript.

L 26-27: in my opinion, after “by-product” (too generic), I would add “Palm Kernel Cake”. Authors are also invited to consider the concept of circular economy (see commentary on lines 30-35). Thanks.

L 30: I find the use of "in this context" more effective than "in this sense". Thanks.

L 30-35: recovery and valorization of agro-industrial residues are currently indicated as key factors for the development of the circular economy models and to promote the environmental sustainability of production systems. International scientific opinion is very interested in these issues, so I would suggest to the authors, before mentioning the economic aspects related to the use of by-products in ruminant feeding, to recall them briefly. In this regard, I suggest using manuscript doi:10.3390/ani9110918 as a template (especially the introduction), which I strongly recommend using as a reference. Thanks.

L 35: I suppose the authors referring to “feeds” instead that “foods”. Please check. Thanks.

L 39 (and throughout the manuscript): with regards to "ideal inclusion level" see the replacing option highlighted on lines 13-14. Thanks.

L 51-52: please paraphrase the sentence excluding "to keep in mind" (a very non-technical expression in my opinion). Thanks.

L 68-70: I can assume that of the 3 goats used to test each diet, one per group was Anglo Nubian. However, this was not made explicit in the text.

L 81 (and throughout the manuscript): please use “0X: 00 h” as the time format. Thanks.

L 111-112: the authors state (L 107) that behavioral observations were conducted for 24 hours, recording the behavior of the goats based on 5-minute intervals. Based on my expertise, I believe that the authors "captured" the goat’s behavior by using the time budget technique combined with the scan sampling (missing the reference list it is difficult for me to evaluate the bibliography mentioned by the authors, so I can only guess!). Well, the authors refer to observation intervals of 2 hours (..."These data were recorded by 8 trained evaluators, distributed in pairs during time intervals of 2 h" ...). It is therefore not clear whether each pair of observers recorded the animal behavior for only 2 hours (and not for 24 as mentioned above), or if in the 24 hours of observation the behavior of the animals was synthesized on scan times of 2 hours rather than 5 minutes. Authors are invited to clarify, thanks.

L 156-166: consistently to the title suggestions (see line 3), the nitrogen balance (paragraph 2.8) and the assessment of blood concentration of proteins and urea (2.7. Blood metabolites) could be unified in a single paragraph as "Nitrogen metabolism". Thanks.

L 191-193: in my opinion, the model with which the regression equations reported in the text were obtained, as well as the included variables, should be described. Thanks.

L 194-195: Did the authors observe any diet-by-period interactions?

L 198 (and throughout the manuscript): according to the journal’s template, the p-value should be reported in italic and uppercase. Thanks.

L 204 (Table 2): I realize that it might seem obvious, but no indication as to how and when the animals were weighed is reported in the materials and methods section. Authors are requested to update these. Thanks.

L 206 (and for other tables): according to the journal template, acronyms should be specified at the first mention, just as tables should be self-explanatory. Therefore, the abbreviations reported in the regression formulas at the foot of the tables (e.g., FBW, DMI, etc.), if not already specified in the text, should be explained, as well as the variables (X) reported in the regression formulas. Thanks.

Regarding the regression equations, I would like to point out to the authors that the placement at the bottom of the table diminishes the value of the equations (as if they were simple notations!). Since a high value of the coefficient of determination is reported for many equations, it might be useful, in my opinion, to tabulate the regressions obtained by reporting the significant ones. The following manuscript doi.org/10.3168/jds.S0022-0302(99)75525-6 is, in my opinion, a useful draft.

L 230: please, see the comments reported for lines 156-166. Thanks.

L 247: authors are advised to check whether discussions can be "broken up" into paragraphs. Thanks.

L 252: the authors assert that the amount of PKC in feed refusals increases as the level of inclusion of the byproduct in the diet increases. On what evidence do they claim this? Did they use an ingredients separator device to isolate the residual PKC? In the materials and methods section, I have not found any description of a similar approach. Also, I guess the PKC was provided milled, so it might be interesting to know how the authors appreciated, if not visually, the residual amount of PKC. Thanks.

L 256: as first mentioned, the CNF abbreviation should be explained. Thanks.

L 280-281: Please, see the comments reported for line 252. Thanks.

L 328: Please, to avoid redundancies, replace “fed feed” with “provided feed”. Thanks.

L 351-360: I agree with the authors' explanation provided with regard to the blood urea trend. Nonetheless, I wanted to point out that this discussion can be further enriched in my opinion by considering not only the CP intake but also the rate and the extent of protein degradability in the rumen and the relative proportion with the NSC. Although the authors have not tabled the dietary content of soluble protein and NPN, I believe that these aspects, even if only a speculative option, can be approached. In this regard, the following manuscript doi.org/10.3390/ani10030515 can be used as a useful draft, that I recommended as a reference. Thanks.

Author Response

Comments to the author

Correction and answers

Reviewer-3

General comments

Dear authors,

I have evaluated your manuscript (ID animals 1882155) concerning the effects of including palm kernel cake in the diet of dairy goats. I believe the manuscript is fascinating since it provides useful information for the "safe" reuse of palm kernel cake in dairy goats' diets, for which little information is available as far as I know. Therefore, the research's contribution to the literature could be greatly appreciated. Nevertheless, in my opinion, the manuscript deserves to be revised in different parts. In addition, bibliographic references have been omitted, making the revision work very difficult and, at least partially, incomplete. So, I suggested a major revision. My suggestions are detailed below, section by section. Hoping to have contributed to improving the manuscript quality. Good works.

Dear reviewer, thank you very much for your time, attention and contributions to improve our manuscript.

Specific comments

L3: I believe that the title excludes an important part of the research (the N balance) while it provides a partially true indication since the metabolic profile (generally referred at blood) concerns exclusively some blood markers of the N cycle. To point out that the metabolic profile has been studied only regarding N metabolism and that, at the same time, the N balance between treatments has been measured, I suggest replacing "Metabolic profile" (considered too generic in this case) with "Nitrogen Metabolism". This generic term, in my opinion, can be fruitfully used to summarize both the effects on plasma metabolites and the N balance as a whole. Thanks.

Thank you very much for your suggestion. Dear reviewer, the change was made according to you suggestion.

L 13-14: I suggest avoiding the adjective "ideal" (too pompous), which can be replaced by "optimal inclusion rate" that fits better the study purposes, in my opinion. Furthermore, it would be advisable to specify in which farming system the palm kernel cake was used or, more simply, in which type of basal diet it's incorporated. In my opinion, the emphasis on this detail is important, as the trial was performed on "feedlot" goats fed with a silage-based diet, which results in both unusual conditions for goat farming in large parts of the world. Thanks.

Thank you very much for your suggestion. Dear reviewer, the “ideal inclusion level” was changed by “optimal inclusion rate” as suggested. Furthermore, we changed "inclusion rate" instead of "inclusion level" throughout the text.

Dear reviewer, we did not understand the second part of this suggestion. We add the name of the company where we buy the palm kernel cake. The description of the diet was in table 1; however, another reviewer suggested that we delete this table because it is the same as in the previous published manuscript. The diet is explained in the text and in the table of the previously published manuscript.

L 15 (and throughout the manuscript): see the comment regarding line 3 about the replacement of metabolic profile with nitrogen metabolism. Thanks.

Thank you very much for your suggestion. Dear reviewer, the change was made according to you suggestion.

L 16: I apologize for the pedantry, but information such as body weight and days in milk is probably too superfluous in this part of the manuscript.

Thank you very much for your suggestion. Dear reviewer, we don't think your comment is pedantic because your effort is to improve our manuscript. On the other hand, we think that this information is important for the presentation letter of the manuscript, the abstract. Please, we ask you to let us keep this data. However, if your recommendation in a new review is to remove; We will remove this text.

L 26-27: in my opinion, after “by-product” (too generic), I would add “Palm Kernel Cake”. Authors are also invited to consider the concept of circular economy (see commentary on lines 30-35). Thanks.

Thank you very much for your suggestion. Dear reviewer, we added “palm kernel cake” as you suggested. However, we add this word as an individual keyword. We understand that “by-product” is generic but we consider that this is the objective of the keywords. Increase and specify the search area for researchers to find our study. With both keywords, it will be easier for authors to find our study.

L 30: I find the use of "in this context" more effective than "in this sense". Thanks.

Thank you very much for your suggestion. Dear reviewer, we made the change according to your suggestion.

L 30-35: recovery and valorization of agro-industrial residues are currently indicated as key factors for the development of the circular economy models and to promote the environmental sustainability of production systems. International scientific opinion is very interested in these issues, so I would suggest to the authors, before mentioning the economic aspects related to the use of by-products in ruminant feeding, to recall them briefly. In this regard, I suggest using manuscript doi:10.3390/ani9110918 as a template (especially the introduction), which I strongly recommend using as a reference. Thanks.

Thank you very much for your suggestion. Dear reviewer, we added information in the introduction after reading the proposed manuscript according to your suggestion.

L 35: I suppose the authors referring to “feeds” instead that “foods”. Please check. Thanks.

Thank you very much for your observation. Dear reviewer, we made the change according to your suggestion.

L 39 (and throughout the manuscript): with regards to "ideal inclusion level" see the replacing option highlighted on lines 13-14. Thanks.

Thank you very much for your suggestion. Dear reviewer, the “ideal inclusion level” was changed by “optimal inclusion rate” as suggested. Furthermore, we changed "inclusion rate" instead of "inclusion level" throughout the text.

L 51-52: please paraphrase the sentence excluding "to keep in mind" (a very non-technical expression in my opinion). Thanks.

Thank you very much for your observation. Dear reviewer, we made the change according to your suggestion.

L 68-70: I can assume that of the 3 goats used to test each diet, one per group was Anglo Nubian. However, this was not made explicit in the text.

Thank you very much for your observation. Dear reviewer, we made changes to explain better the Latin Square. We use three parallel Latin Squares; each Square used four animals of the same breed (This distribution to include the effect of breed in the Statistical Analysis and avoid its influence), four periods, and four treatments. This makes a total of 47 degrees of freedom.

L 81 (and throughout the manuscript): please use “0X: 00 h” as the time format. Thanks.

Thank you very much for your observation. Dear reviewer, we made the change according to your suggestion here and throughout the manuscript.

L 111-112: the authors state (L 107) that behavioral observations were conducted for 24 hours, recording the behavior of the goats based on 5-minute intervals. Based on my expertise, I believe that the authors "captured" the goat’s behavior by using the time budget technique combined with the scan sampling (missing the reference list it is difficult for me to evaluate the bibliography mentioned by the authors, so I can only guess!). Well, the authors refer to observation intervals of 2 hours (..."These data were recorded by 8 trained evaluators, distributed in pairs during time intervals of 2 h" ...). It is therefore not clear whether each pair of observers recorded the animal behavior for only 2 hours (and not for 24 as mentioned above), or if in the 24 hours of observation the behavior of the animals was synthesized on scan times of 2 hours rather than 5 minutes. Authors are invited to clarify, thanks.

Thank you very much for your observation. Dear reviewer, we made the change according to your suggestion. The feeding behavior was analyzed for 24 h, in intervals of 5 min. The data were recorded by 8 trained evaluators, distributed in pairs during time intervals of 2 h. During these 2 hours, the two evaluators performed feeding behavior every 5 minutes; after this time, two new evaluators performed the same exercise for the same amount of time.

L 156-166: consistently to the title suggestions (see line 3), the nitrogen balance (paragraph 2.8) and the assessment of blood concentration of proteins and urea (2.7. Blood metabolites) could be unified in a single paragraph as "Nitrogen metabolism". Thanks.

Thank you very much for your observation. Dear reviewer, we made the change according to your suggestion.

L 191-193: in my opinion, the model with which the regression equations reported in the text were obtained, as well as the included variables, should be described. Thanks.

Thank you very much for your observation. Dear reviewer, the detail to obtain the regression equations is described: Orthogonal Polynomial Contrasts to determine the linear (-3 -1 +1 +3) and quadratic (+1 -1 -1 +1) effects. The variables were added to the text.

L 194-195: Did the authors observe any diet-by-period interactions?

Thank you very much for your observation. Dear reviewer, this information is not included here because no interactions between diet-by-period were observed in the complete analysis.

L 198 (and throughout the manuscript): according to the journal’s template, the p-value should be reported in italic and uppercase. Thanks.

Thank you very much for your observation. Dear reviewer, the changes were made according to your suggestion.

L 204 (Table 2): I realize that it might seem obvious, but no indication as to how and when the animals were weighed is reported in the materials and methods section. Authors are requested to update these. Thanks.

Thank you very much for your observation. Dear reviewer, the information was added in the material and methods subject after the milking management.

L 206 (and for other tables): according to the journal template, acronyms should be specified at the first mention, just as tables should be self-explanatory. Therefore, the abbreviations reported in the regression formulas at the foot of the tables (e.g., FBW, DMI, etc.), if not already specified in the text, should be explained, as well as the variables (X) reported in the regression formulas. Thanks.

Thank you very much for your observation. Dear reviewer, a new table (Table 6) without acronyms (Except for the NDFpd described as a footnote) and with the regression equations found was added.

Regarding the regression equations, I would like to point out to the authors that the placement at the bottom of the table diminishes the value of the equations (as if they were simple notations!). Since a high value of the coefficient of determination is reported for many equations, it might be useful, in my opinion, to tabulate the regressions obtained by reporting the significant ones. The following manuscript doi.org/10.3168/jds.S0022-0302(99)75525-6 is, in my opinion, a useful draft.

Thank you very much for your observation. Dear reviewer, a new table (Table 6) without acronyms (Except for the NDFpd described as a footnote) and with the regression equations found was added.

L 230: please, see the comments reported for lines 156-166. Thanks.

Thank you very much for your observation. Dear reviewer, we agree with the changes in the title and in the objective of the manuscript; however, we ask that you let us keep this description (nitrogen balance and blood metabolites) because some blood metabolites do not participate solely in nitrogen metabolism. If in a new revision you ask us to join these subjects, we will do it according to your suggestion.

L 247: authors are advised to check whether discussions can be "broken up" into paragraphs. Thanks.

Thank you very much for your observation. Dear reviewer, we searched the Instructions to Authors and found no limitations. Our previously published manuscript was divided into paragraphs. We will contact the editorial board to ask about this point.

L 252: the authors assert that the amount of PKC in feed refusals increases as the level of inclusion of the byproduct in the diet increases. On what evidence do they claim this? Did they use an ingredients separator device to isolate the residual PKC? In the materials and methods section, I have not found any description of a similar approach. Also, I guess the PKC was provided milled, so it might be interesting to know how the authors appreciated, if not visually, the residual amount of PKC. Thanks.

Thank you very much for your observation. Dear reviewer, we understand that it is hard to believe. How do goats separate dry ingredients from wet silage? We don't know, but the refusals contained quite a bit of PKC. This behavior was also observed in another experiment developed by our study group (manuscript cited Silva et al., 2021).

Unfortunately, we did not develop an analysis to show this behavior. Furthermore, we lost the photographs of these refusals when our camera was stolen from our laboratory facilities. PKC particles are visible, similar to soybean meal particles but different in color.

L 256: as first mentioned, the CNF abbreviation should be explained. Thanks.

Thank you very much for your observation. Dear reviewer, we are wrong with this abbreviation. This word is actually NFC, non-fibrous carbohydrates.

L 280-281: Please, see the comments reported for line 252. Thanks.

Thank you very much for your observation. Dear reviewer, as described before, this theory is only visual. We hope you understand our limitations.

L 328: Please, to avoid redundancies, replace “fed feed” with “provided feed”. Thanks.

Thank you very much for your observation. Dear reviewer, the change was made according to your suggestion.

L 351-360: I agree with the authors' explanation provided with regard to the blood urea trend. Nonetheless, I wanted to point out that this discussion can be further enriched in my opinion by considering not only the CP intake but also the rate and the extent of protein degradability in the rumen and the relative proportion with the NSC. Although the authors have not tabled the dietary content of soluble protein and NPN, I believe that these aspects, even if only a speculative option, can be approached. In this regard, the following manuscript doi.org/10.3390/ani10030515 can be used as a useful draft, that I recommended as a reference. Thanks.

Thank you very much for your observation. Dear reviewer, we read the manuscript and did not find a way to use this theory because the serum urea found was within the reference values observed in lactating goats. However, we cite the manuscript as a complement to the metabolism of nitrogen to urea.

Round 2

Reviewer 1 Report

Dear authors thanks for the revised version of your manuscript here are my concerns:

I strongly advise the authors to remove the data of the intake and the milk composition from the tables in this article, as you already used the same data in your previous published work, you can refer to these data in your introduction and the new parameters will be the novelty in this article, i mean that you already got some promising results from your previous work on intake and milk production, but still the effects on the parameters that you did in your current study are unknown, so you did this study. So please remove any published data in this manuscript.

Please write all numbers in the tables in three digits (e.g., xxx, x.xx, 0.xx, 0.0x, ....)

Author Response

Dear reviewer, thank you very much for your patience, expertise, time, attention, and contributions to improve our manuscript. Improving this manuscript was possible thanks to your contribution.

Based on your suggestion, we removed the composition data for the milk. However, we ask that you allow us to keep your intake data (Dry Matter and Crude Protein). These data are very important to explain some results. Furthermore, eliminating these data (Intake of Dry Matter and Crude Protein) would leave the consumption table incomplete. The authors would need to search the other manuscript to find these parameters to understand other outcomes (eg intake of other nutrients) that may be explained by DM and CP intake.

Furthermore, we cite the published manuscript by repeating the diet composition table, and in this sense, the authors may think that we are practically and exaggeratedly trying to force them to read the published manuscript.

To avoid duplication, we initially didn't just modify the display units; we also used more statistical pressure, which resulted in relatively different values than in the previously published article. We hope you can understand our point of view.

We complete the numbers in tables in three digits. However, we follow a line of thought about keeping three digits as the minimum number of digits because some parameters have a larger number of digits that cannot be reduced to three digits. We hope you can understand our point of view.

Reviewer 2 Report

The revised manuscript has been improved. I have no further comments

Author Response

Dear reviewer, thank you very much for your patience, expertise, time, attention, and contributions to improve our manuscript. Improving this manuscript was possible thanks to your contribution.

Reviewer 3 Report

Dear authors,
I have revised the new version of your manuscript (ID animals-1882155). First, I would like to thank you for the patience and expertise with which you went through the review process. Regarding the manuscript, I have no other comments to make. In addition, I believe that what the authors requested regarding my previous comments for lines 13-14; 16, 230, and 247 can be maintained. I hope to see your manuscript published soon.
Best wishes.

Author Response

(The authors gave the same response as above.)
